# Association Studies on Gut and Lung Microbiomes in Patients with Lung Adenocarcinoma

**DOI:** 10.3390/microorganisms11030546

**Published:** 2023-02-21

**Authors:** Yaqiong Guo, Wenjie Yuan, Na Lyu, Yuanlong Pan, Xiaoqing Cao, Yuxuan Wang, Yi Han, Baoli Zhu

**Affiliations:** 1CAS Key Laboratory of Pathogenic Microbiology and Immunology, Institute of Microbiology, Chinese Academy of Sciences, Beijing 100101, China; 2University of Chinese Academy of Sciences, Beijing 100049, China; 3Beijing Tuberculosis and Thoracic Tumor Research Institute, Beijing 101149, China; 4Beijing Chest Hospital, Capital Medical University, Beijing 101149, China; 5Jinan Microecological Biomedicine Shandong Laboratory, Jinan 250117, China; 6Department of Pathogenic Biology, School of Basic Medical Sciences, Southwest Medical University, Luzhou 646000, China

**Keywords:** gut microbiome, lung microbiome, lung adenocarcinoma, gut–lung axis

## Abstract

Lung adenocarcinoma (LADC) is a prevalent type of lung cancer that is associated with lung and gut microbiota. However, the interactions between these microbiota and cancer development remain unclear. In this study, a microbiome study was performed on paired fecal and bronchoalveolar lavage fluid (BALF) samples from 42 patients with LADC and 64 healthy controls using 16S rRNA gene amplicon and shotgun metagenome sequencing, aiming to correlate the lung and gut microbiota with LADC. Patients with LADC had reduced α-diversity in the gut microbiome and altered β-diversity compared with healthy controls, and the abundances of *Flavonifractor*, *Eggerthella*, and *Clostridium* were higher in the gut microbiome of LADC patients. The increased abundance of microbial species, such as *Flavonifractor plautii*, was associated with advanced-stage LADC and a higher metastasis rate. Phylogenetically, *Haemophilus parainfluenzae* was the most frequently shared taxon in the lung and gut microbiota of LADC patients. Gut microbiome functional pathways involving leucine, propanoate, and fatty acids were associated with LADC progression. In conclusion, the low diversity of the gut microbiota and the presence of *H. parainfluenzae* in gut and lung microbiota were linked to LADC development, while an increased abundance of *F. plautii* and the enriched metabolic pathways could be associated with the progression of LADC.

## 1. Introduction

Lung cancer is one of the leading causes of cancer death worldwide, and LADC is the most prevalent subtype of lung cancer, accounting for approximately 40% of all cases of lung cancer [1]. In the search for better diagnostic tools and new therapeutic regimens, many studies have been performed to identify the role of gut microbiota in association with lung cancer development. Recently, it has been reported that some gut bacterial groups, such as *Bacteroides* and *Ruminococcus*, from the gut microbiota of lung cancer patients were associated with early-stage lung cancer [2]. Another study of gut samples of patients with LADC showed that increased abundances of *Bacteroides*, *Pseudomonas*, and *Ruminococcus* were associated with LADC development [3]. Dysbiosis of the lung microbiome has also been associated with lung cancer patients; the BALF was used to represent the lung microbiome and revealed that the *Veillonella* and *Megasphaera* were significantly increased in patients with lung cancer [4]. Another study using lung cancer tissues showed that the bacterial genera *Romboutsia*, *Novosphingobium*, and *Acinetobacter* were enriched compared with the adjacent normal tissues [5]. A mouse model provided evidence that lung microbiota could provoke inflammation associated with LADC by activating lung-resident γδ T cells, which could promote neutrophil infiltration and tumor cell proliferation [6].

Based on the association studies of gut and lung microbiota with cancer development, the “gut–lung axis” was proposed to interpret the crosstalk between the gut and lung [7,8]. A recent study using sputum and fecal samples to analyze the link between gut and lung microbiota showed that both contained the genus *Streptococcus* in patients with lung cancer [9]. However, the microbial community of sputum samples is commonly contaminated with oral bacteria [10,11,12], so it may not be the optimal sample type to study the lung microbiome associated with lung cancer. Nevertheless, how the gut and lung microbiomes affect the occurrence of lung cancer through the gut–lung axis does need further investigation.

This study focused on gut and lung microbiomes in association with LADC, using both the fecal and BALF samples to analyze the bacterial species that were correlated with the development of LADC. The shared bacterial components of fecal and BALF microbiota, and the metabolic pathways of gut microbiota associated with LADC development, were also analyzed.

## 2. Materials and Methods

### 2.1. Study Cohort and Sample Collection

A total of 43 patients with LADC and 64 healthy controls met the inclusion criteria and were included in the final study. Progression-free survival (PFS) was determined for 32 patients, excluding those who were lost to follow-up. Patients were diagnosed with LADC according to the TNM staging system (8th edition) [13]. The demographic and clinical characteristics of the study participants that were captured at baseline included sex, age, and smoking status. For patients with LADC, other clinical and pathological features, including disease stage and distant metastasis, were also captured. To be included in this study, patients met the following criteria: (1) aged 18–80 years; (2) the LADC was diagnosed by pathological or cytological findings. The exclusion criteria were as follows: (1) having a history of malignancy; (2) having a history of chemotherapy, radiotherapy, or cancer surgery; and (3) having received probiotics or antibiotics within the previous 1 month before enrollment. All healthy participants were the spouse or family members of the hospital patients.

A total of 42 fecal samples with paired BALF were collected from the patients with LADC, and 64 fecal samples were collected from the healthy controls. In another patient with LADC, only a stool sample and no BALF sample was collected. All fecal and BALF samples were obtained before the patients received treatment. The BALF was acquired with sterile isotonic saline in order to obtain the lesion site. All samples were immediately refrigerated and stored at −80 °C for further analysis.

### 2.2. 16S rRNA Gene Amplicon Sequencing

Bacterial DNA was extracted from the fecal and BALF samples using the QIAamp PowerFecal DNA Kit (Qiagen) according to the manufacturer’s instructions. The V3–V4 regions of the bacterial 16S rRNA genes were amplified using the 341F (5′-CCTACGGGNBGCASCAG-3′) and 805R (5′-GACTACNVGGGTATCTAATCC-3′) primers. High-throughput sequencing was performed in 250-bp paired end reads using the Illumina NovaSeq 6000 platform. The raw data were processed using QIIME2-2020.8 [14]. Low-quality reads were filtered with the QIIME2 plugin q2-dada2 [15]. α-Diversity and β-diversity were analyzed using the q2-diversity plugin. Taxonomy analysis was performed with the q2-feature-classifier plugin [16] and the classify-sklearn naïve Bayes taxonomy classifier against the SILVA database [17].

### 2.3. Shotgun Metagenome Sequencing

Bacterial DNA was isolated from the fecal samples for shotgun metagenomic sequencing. These samples were sequenced in 150-bp paired end reads using the Illumina NovaSeq 6000 platform. Human DNA contamination was removed from the raw reads and low-quality sequences were trimmed using KneadData v0.7.2 software (https://huttenhower.sph.harvard.edu/kneaddata/, accessed on 21 June 2019). The metagenomic reads were assembled into contigs individually for each sample using MEGAHIT v1.2.4-beta [18]. Gene prediction was performed with Prodigal v2.6.3 [19]. A non-redundant gene catalog was constructed by clustering predicted genes based on sequence similarity at 95% identity and 90% coverage of the shorter sequence using CD-HIT v4.8.1 [19]. The metagenomic reads coverage of each sample was calculated using BBMap v38.57 (https://github.com/BioInfoTools/BBMap, accessed on 11 July 2019) with the non-redundant gene catalog as the reference. MetaPhlAn3 v3.0.13 [20] was used to perform taxonomic classification and analysis by mapping metagenomic reads against a library of clade-specific markers. Differentially enriched functional pathways in the Kyoto Encyclopedia of Genes and Genomes (KEGG) database were identified according to Welch’s *t*-test using STAMP [21,22]. Virulence factors were analyzed according to the virulence factor database (VFDB) [23]. Genes in the gene catalog were identified as these virulence factors using BlastP (identity > 80%, coverage > 80%), and their relative abundances were determined accordingly.

### 2.4. Phylogenetic Analysis between Gut and Lung Microbiomes

After 16S rRNA gene sequencing analysis, shared genera in the fecal and paired BALF samples from the same patient were identified. To further explore the shared species, characteristic sequences of the shared genera were extracted for a comparative analysis of phylogenetic trees based on FastTree 2.1.10 software [24]. A multiple sequence alignment was performed by MAFFT v7.505 software. To determine the specific shared species, the National Center for Biotechnology Information (NCBI) nucleotide BLAST software was used for alignments of the characteristic sequences against the nucleotide collection (nr/nt) databases with default parameters. The shared species were identified by combining the comparative analysis of phylogenetic trees and the highest identity values (>97%) of the BLAST alignment results.

### 2.5. Statistical Analysis

Analyses of α-diversity and β-diversity were conducted using the vegan v2.5-7 package [25]. α-Diversity was estimated by the Shannon index, evenness index, and observed features, while β-diversity analysis was conducted using the unweighted UniFrac distance and the Bray–Curtis distance matrix. Statistical significance analyses were performed using an ANOSIM test in PCoA analyses. The Galaxy online platform (http://huttenhower.sph.harvard.edu/galaxy/, accessed on 26 March 2021) was used to determine the taxa of biomarkers using the linear discriminant analysis (LDA) effect size (LEfSe). Fisher’s exact test was performed for categorical variables, and Student’s *t*-test was used for continuous variables to compare clinical parameters. The nonparametric Wilcoxon rank-sum test was used for comparison between the two groups. Statistically significant differences between groups of the KEGG pathways were determined by Welch’s *t* test using STAMP.

## 3. Results

### 3.1. Characteristics of Study Participants

A total of 107 participants who met the inclusion criteria were enrolled in the study. Among the participants, 43 were newly diagnosed with LADC and had not previously received any anticancer therapy, nor been treated with antibiotics in the month prior to sample collection. The 64 healthy individuals were healthy family members of the patients with LADC and were used as controls. The patients and healthy controls were similar with respect to sex and smoking status. Fecal samples were collected from the healthy controls and all patients with LADC. The BALF was also collected from 42 of the 43 patients with LADC. The patients with LADC were all first-time diagnosed with different stages of LADC, including stage I (46.5%), stage II (7.0%), stage III (23.3%), and stage IV (23.3%). Most of these patients (76.7%) had non-metastatic lung cancer (Table 1).

### 3.2. The Gut Microbiota of Patients with LADC Exhibited Reduced α-Diversity and Altered Bacterial Composition

Amplicon sequencing of the V3–V4 hypervariable regions of the 16S rRNA gene and metagenome sequencing were performed for all fecal samples to investigate the bacterial composition of the gut microbiota. For 16S rRNA gene sequencing, a total of 20,940,816 raw reads were acquired from 107 gut samples, with an average of 195,709 reads per sample (43 patients with LADC had a total of 7,682,042 reads and an average of 178,652 reads per sample; 64 healthy people had a total of 13,258,774 reads and an average of 207,168 reads per sample) (Appendix A). Sequences of the 16S rRNA gene V3–V4 region for all 107 fecal microbiota samples were assigned to 291 genera, and the relative abundance and diversity were assessed and compared between patients and healthy individuals using Shannon indexes, Evenness indexes, and Observed features (Figure 1A). α-Diversity was reduced in patients with LADC, and this could potentially influence the disease progression and severity [26]. A principal coordinates analysis (PCoA) of the composition and abundance of the gut microbiota showed that the β-diversity of the patients with LADC was clearly separated from that of the healthy group (unweighted Unifrac: R = 0.1152, *p* = 0.001; Bray–Curtis: R = 0.1104, *p* = 0.002) (Figure 1B). This indicated that the bacterial composition of the gut microbiota of the healthy group was significantly different from that of the patients with LADC.

*Firmicutes* and *Bacteroidota* were the most abundant phyla in the gut microbiota of patients with LADC and healthy controls, with *Firmicutes* slightly reduced and *Bacteroidota* tending to be increased in LADC compared with the abundances for healthy controls (Appendix A). At the genus level, *Bacteroides* was the most abundance taxa in the gut samples of patients with LADC. A linear discriminant analysis effect size (LEfSe) was used to generate a histogram to present the differences in bacterial abundance between the patients with LADC and healthy controls. Some of the bacterial taxa were consistent with previous reports. At the genus level, *Coprococcus* [27], *Dorea* [28], *Lachospiraceae* UCG-001, *Ruminococcus torques group*, *Eubacterium hallii group* [28], and *Dialister* [29,30] were enriched in the healthy group, while *Prevotella* [27,29], *Enterococcus* [3,28], *Flavonifractor*, *Clostridium innocuum group*, and *Eggerthella* [9] were enriched in the LADC group (Figure 2A). To analyze the differences in the bacterial composition at the species level, metagenome sequencing was employed for both groups of participants (healthy controls and patients with LADC). A total of 7,966,458,130 raw reads were acquired from 107 gut samples, with an average of 74,452,800 reads per sample (43 patients with LADC had a total of 3,242,831,028 reads and an average of 75,414,675 reads per sample; 64 healthy people had a total of 4,723,627,102 reads and an average of 73,806,673 reads per sample) (Appendix A). The species *Bacteroides plebeius*, *Dialister* sp. *CAG-357*, *Eubacterium* sp. *CAG-38*, *Roseburia intestinalis*, and *Ruminococcus torques* were enriched in healthy controls, while *Bacteroides thetaiotaomicron*, *Fusobacterium mortiferum*, *Flavonifractor plautii*, *Enterococcus gallinarum*, and *Eggerthella lenta* were enriched in the patients with LADC (Appendix A). The consistency of the taxonomy annotation by 16S rRNA gene sequencing and metagenome sequencing was confirmed by the observation that the top ten genera in relative abundance were all the same for the two sequencing methods (Appendix A). Several pathogenic bacteria were enriched in the gut microbiota of patients with LADC (Figure 2A and Appendix A), including *Enterococcus faecium* [31], *Enterococcus avium* [32], *Clostridium innocuum* [33], and *Eggerthella lenta* [34]. The dysbiosis of gut microbiota makes an individual more vulnerable to cancer, as pathogens can exert negative effects on the host’s physiology, metabolism, and immune system. Meanwhile, short-chain fatty acid-producing bacteria, such as *Coprococcus*, *Ruminococcus*, *Eubacterium hallii*, and *Roseburia intestinalis*, were decreased in the LADC group [35,36].

Next, the gut bacteria associated with tumor stages and metastasis were compared, with tumor stages I and II comprising the early-stage group, and stages III and IV comprising the advanced-stage group. Several lung cancer-related gut microbes, such as *Flavonifractor*, *Eggerthella*, and *Clostridium innocuum group*, increased with cancer progression and the appearance of metastases (Figure 2B,C). Concurrently, the species-level bacteria associated with cancer progression were also revealed and included *Flavonifractor plautii*, *Eggerthella lenta*, *Clostridium innocuum*, and *Clostridium aldenense* (Figure 3A,B). To investigate the impact of gut microbial diversity on LADC progression, the relationship of α-diversity and PFS was tested by stratifying patients into high versus low categories based on the median of the evenness index in the gut microbiome. This analysis demonstrated that patients with high α-diversity tended to exhibit prolonged PFS versus those patients with a low diversity, but the difference was not significant (Figure 2D).

### 3.3. Bacterial Species Found in Both Gut and Lung Microbiota of Patients with LADC

To explore whether there were shared bacterial taxa between the gut and lung microbiota of patients with LADC, a comparative analysis of the 16S rRNA gene sequencing data of the BALF samples (which represent lung microbiota) and of the gut microbiota was performed. A total of 6,161,611 raw reads were acquired from 42 BALF samples, with an average of 146,705 reads per sample (Appendix A). At the phylum level, *Bacteroidota* was the most abundant taxa, followed by *Firmicutes* in the BALF samples (Appendix A), which was opposite to that of the gut samples. At the genus level, *Prevotella* and *Alloprevotella* were the most abundant taxa in the BALF samples of LADC patients (Figure 4), which was different from the gut microbiota of these patients, where the genus *Bacteroides* was the most abundant taxa. The abundance of *Leptotrichia* and *Fusobacterium* in the lung microbiome tended to increase in patients with advanced LADC, indicating that these taxa could be involved in lung cancer progression.

A total of 94 genera appeared in both fecal and BALF samples based on taxonomy annotations using the QIIME2 plugin classify-sklearn. At the individual level, the genus *Streptococcus* was present in the paired lung and gut samples of 26 patients, followed by *Prevotella* in the paired lung and gut samples of 23 patients, *Fusobacterium* in 13 patients, *Haemophilus* in 10 patients, and *Bacteroides* in 10 patients (Appendix A). To further explore the shared taxa at the species level, FastTree 2.1.10 software was used to separately reconstruct bacterial phylogenetic trees for lung and gut microbiota based on 16S rRNA gene sequences from all 5 of the above genera, and blastn of the NCBI was used to annotate these 16S rRNA gene sequences at the species level. For the genus *Streptococcus*, the 144 16S rRNA gene sequences were grouped in 10 species, of which *Streptococcus salivarius* was detected in both the gut and lung samples of 5 patients (Figure 5A, Appendix A). For *Fusobacterium*, the 148 sequences were classified into 5 species, and the species *Fusobacterium mortiferum* was shared in the gut and lung samples of one patient (Figure 5B, Appendix A). For *Haemophilus*, the 32 16S rRNA gene sequences were classified into 3 species, and *Haemophilus parainfluenzae* was the shared species in the lung and gut microbiota in 10 patients (Figure 5C, Appendix A). For *Bacteroides*, the 391 sequences were grouped in 12 species, 3 of which were detected in paired lung and gut samples. The shared species were *Bacteroides fragilis* in 1 patient, *Bacteroides uniformis* in 1 patient, and *Bacteroides stercoris* in 1 patient (Figure 5D, Appendix A). In total, 6 shared species (species detected in paired gut and lung samples) were found. Among them, *Fusobacterium mortiferum* and *H. parainfluenzae* were significantly enriched in gut samples from patients with LADC using metagenomic sequencing data analysis (Appendix A). However, although a total of 6 shared species were detected, there were fewer patients who shared these species in the paired gut and lung samples of LADC patients. *H. parainfluenzae* was the most prevalent shared species, found in both the lung and gut microbiota of 10 patients, which may indicate that *H. parainfluenzae* is a key species in the development of LADC.

For the genus *Prevotella*, the 828 16S rRNA gene sequences were classified into 21 species (Appendix A), and the relative abundance of *Prevotella* was significantly enriched in the gut samples of patients with LADC compared with the healthy controls (Figure 2A). However, no species of the genus *Prevotella* were detected in paired lung and gut samples (shared species).

### 3.4. Comparative Analysis of Metabolic Pathways of Gut Microbiota in Patients with LADC

To explore the functional metabolic pathways in the gut microbiota of patients with LADC, the relative abundance of KEGG pathways between patients with LADC and healthy controls were compared. A non-redundant gene catalog based on metagenome sequencing data (10.4 Gb per sample) was constructed from the gut samples of 43 patients with LADC and 64 healthy controls, and consisted of 3,328,337 unique genes. Metabolic pathways were identified by aligning the whole gene set of the gene catalog against the KEGG database using DIAMOND (v0.9.31.132). There were 18 different pathway modules between patients with LADC and healthy controls. Of these, 9 pathway modules were enriched in patients with LADC and were predominantly involved in carbohydrate metabolism and amino acid metabolism. In total, 9 pathway modules were enriched in healthy controls, and these were mainly involved in carbohydrate metabolism and the metabolism of cofactors and vitamins (Figure 6). Among the related metabolic modules of gut microbiota affecting the immune system and inflammatory response, it has been reported that leucine can activate the mammalian target of the rapamycin (mTOR) signaling pathway in intestinal epithelial cells [37]. Propanoyl-CoA catabolism is a module of the propanoate metabolism. Propanoyl-CoA is metabolized to methylmalonyl-CoA by propionyl-CoA carboxylase [38], and the accumulation of methylmalonyl-CoA has been reported to induce tumor progression [39,40]. In our study, the genes encoding propionyl-CoA carboxylase were enriched in patients with LADC (Appendix A). Acetyl-CoA carboxylase (ACC) is the rate-limiting enzyme of the fatty acid biosynthesis pathway, which has been demonstrated as a potential cancer therapeutic target [41]. The gene encoding ACC had a higher abundance in patients with LADC (Appendix A). In addition, compared with healthy people, the relative abundance of the pathways of thiamine biosynthesis and riboflavin biosynthesis were decreased in the patients with LADC. It is recognized that the gut microbiota is a potential source of B vitamins beyond diet, and thiamine and riboflavin deficiency had been defined as a risk factor for cancer [42,43].

Virulence factor genes in the gut microbiota of all the participants were further analyzed using VFDB [44], and 13 virulence factors had a higher abundance in patients with LADC. These 13 virulence factors belonged to 4 categories: adhesion, extracellular enzymes, survival stress, and immune modulation (Figure 7). Furthermore, among the 13 virulence factor genes, 7 of them were lipopolysaccharide (LPS) biosynthesis genes in the KEGG database (Appendix A) and that are related to host inflammation. LPS is also known to induce the differentiation of Th17 cells in the innate immune system that play a key role in host–pathogen interactions [45,46]. In addition, 10 virulence genes associated with immune modulation were enriched in non-smokers regardless of whether they were in the LADC group or the healthy control group, suggesting that smoking might affect immunity.

## 4. Discussion

In this study, we identified a few taxa of the gut microbiota associated with LADC, and among them, the relative abundances of the genera *Flavonifractor*, *Eggerthella*, and *Clostridium* increased with the tumor stage and metastasis of LADC (Figure 2B,C). Specifically, the species *Flavonifractor plautii*, *Eggerthella lenta*, *Clostridium innocuum*, and *Clostridium aldenense* were associated with the progression of LADC (Figure 3). However, these bacterial taxa were absent in BALF samples from the same patients, which could indicate an indirect effect of gut microbiota on lung cancer through immune responses. For example, *Flavonifractor plautii*, which was reported to be an important species in colorectal cancer development [47], can inhibit the antigen-induced T helper 2 cell (Th2) immune responses [48]. The gut bacteria *Eggerthella lenta* was previously demonstrated to activate Th17 cells and exacerbate colitis in mouse models [49].

The species *H. parainfluenzae* was found in both the lung and gut microbiota of ten patients with LADC. *H. parainfluenzae* was previously reported to be a common bacterium within the healthy oropharynx, but it has never been reported to be present in BALF samples or in the lung microbiota of healthy individuals. *H. parainfluenzae* was described as an opportunistic pathogen that was present in the respiratory tract of patients with chronic obstructive pulmonary disease [50]. We also found that the *H. parainfluenzae* had a higher abundance in the gut microbiome of patients with LADC compared with healthy controls (Appendix A). This suggests that *H. parainfluenzae* is instrumental in LADC development. The genus *Prevotella* was detected in a high frequency in both the gut and lung samples (shared in 23 patients), and was enriched in the gut microbiome of the patients with LADC (Figure 2A). However, owing to the very diversified identity of 16S rRNA gene sequences from the gut and lung microbiota in the same patient (<97%), the role of the genus *Prevotella* in cancer development was uncertain in the current study.

Among the metabolic pathways found to be associated with LADC, we consider that leucine degradation, propanoyl-CoA catabolism, and fatty acid biosynthesis, which all showed significant differences between patients with LADC and healthy controls, are the most important pathways in LADC development. It has been reported that targeting leucine in cancer treatment is effective because leucine can activate the mTOR signaling pathway in intestinal epithelial cells [37], which plays a role in T cell activation [51]. The propionyl-CoA carboxylase enriched in patients with LADC can convert propanoyl-CoA into methylmalonyl-CoA [38], which could induce lung cancer progression [39,40]. The ACC, which regulates fatty acid biosynthesis, was enriched in patients with LADC, indicating that it plays an instrumental role in LADC development. Accordingly, the inhibition of ACC was previously reported to limit tumor growth in mouse models of non-small cell lung cancer (NSCLC) [41]. These results imply that the gut microbiota could link with the lung through the immune system and gut metabolites.

Future work should include a multi-center large population cohort and a verified experiment to evaluate how the gut microbiota is linked to the lung microbiota in lung cancer.

## 5. Conclusions

The presence of the species *H. parainfluenzae* is apparently linked to the development of LADC because it was found in both the lung and gut microbiota of ten patients with LADC, and had a higher abundance in the gut microbiota of patients with LADC. The relative abundance of specific species, such as *Flavonifractor plautii*, *Eggerthella lenta*, *Clostridium innocuum*, and *Clostridium aldenense*, are likely associated with LADC progression as the abundance of these species increased with the progression of cancer development. The pathways of leucine degradation, propanoyl-CoA catabolism, and fatty acid biosynthesis are important factors in LADC progression and indicate that there could be an interaction between the gut and lung through the immune system or gut metabolites.

## Figures and Tables

**Figure 1 microorganisms-11-00546-f001:**
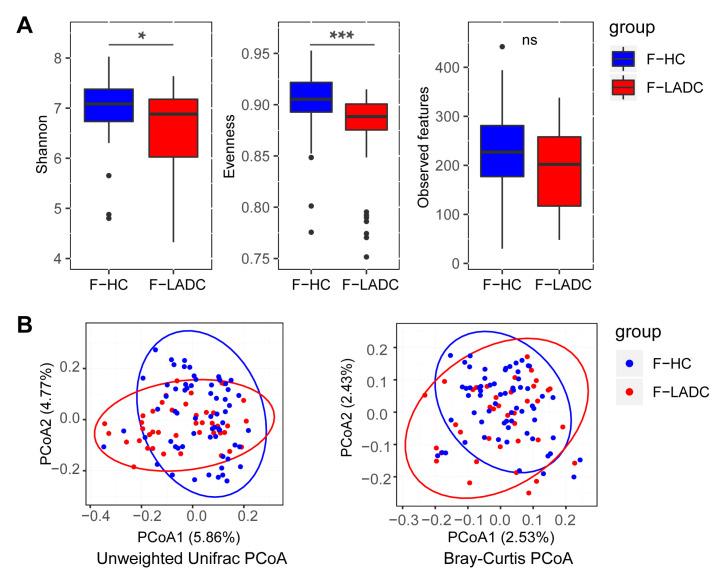
Gut microbiota dysbiosis in patients with LADC. (**A**) Differences in α-diversity between gut samples in LADC group (F-LADC) and healthy control group (F-HC) based on Shannon indexes, Evenness indexes, and Observed features. ***, *p* < 0.001; *, *p* < 0.05; ns, not significant. (**B**) β-Diversity differences between the LADC (red dots) and HC (blue dots) groups were estimated by principal coordinates analysis (PCoA). Left, unweighted UniFrac PCoA; right, Bray–Curtis PCoA. The percentage of variance explained by the first two principal coordinates is indicated in parentheses. Significant differences were observed between patients with LADC and healthy controls with an ANOSIM test (unweighted UniFrac, R = 0.1152, *p* = 0.001; Bray–Curtis, R = 0.1104, *p* = 0.002).

**Figure 2 microorganisms-11-00546-f002:**
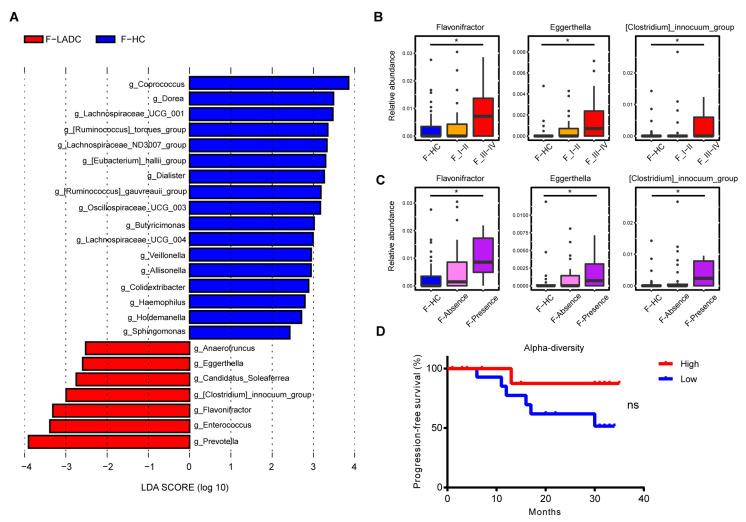
Differences in abundance of gut microbiota in patients with LADC and healthy controls. (**A**) Histogram of the LDA scores, where the LDA score indicates the effective size and ranking of taxa at the genus level (LDA > 2). Red and blue represent the patients with LADC (n = 43) and healthy controls (n = 64), respectively. Three genera were further analyzed in the disease stage (**B**) and distant metastasis (**C**). F-HC, healthy control group (n = 64); F_I-II, early-stage LADC group (n = 23); F_III-IV, advanced-stage LADC group (n = 20); F-Absence, non-metastatic LADC group (n = 33); F-Presence, metastatic LADC group (n = 10); *p*-values were calculated using the two-tailed Wilcoxon rank-sum test: *, *p* < 0.05. (**D**) Survival curves of progression-free survival by evenness index. High (red) vs. low (blue) diversity; log-rank (Mantel–Cox) analysis: ns, not significant.

**Figure 3 microorganisms-11-00546-f003:**
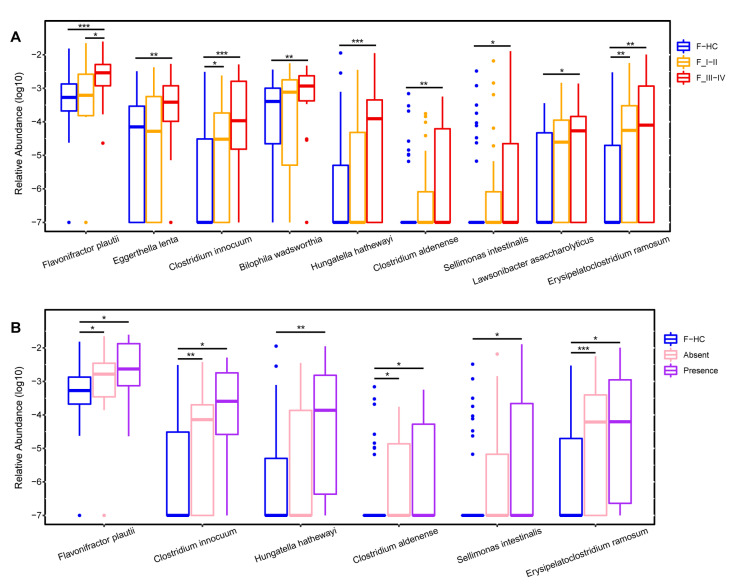
The species significantly increased in the gut microbiota of LADC patients with cancer progression. Boxplots show the relative abundances of species that gradually increased from early-stage to late-stage patients (**A**) and as patients developed distant metastases (**B**) compared with healthy controls. F-HC, healthy control group (n = 64); F_I–II, early-stage LADC group (n = 23); F_III–IV, advanced-stage LADC group (n = 20); F-Absence, non-metastatic LADC group (n = 33); F-Presence, metastatic LADC group (n = 10); *p* values were calculated using the two-tailed Wilcoxon rank-sum test, ***, *p* < 0.001; **, *p* < 0.01; *, *p* < 0.05.

**Figure 4 microorganisms-11-00546-f004:**
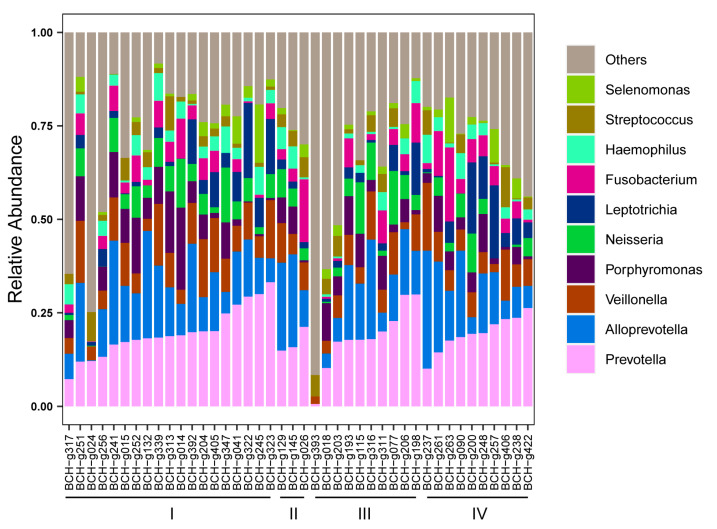
Relative abundance of top 10 genera in the lung microbiota of patients with LADC. The disease staging status in LADC (n = 42) is indicated at the bottom of the figure.

**Figure 5 microorganisms-11-00546-f005:**
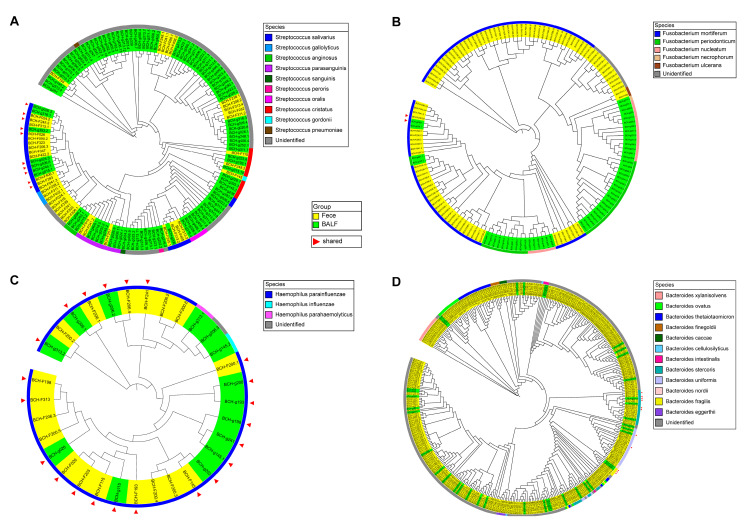
Phylogenetic analysis of the genera *Streptococcus* (**A**), *Fusobacterium* (**B**), *Haemophilus* (**C**), and *Bacteroides* (**D**) with characteristic 16S rRNA gene sequences of paired fecal and BALF samples in patients with LADC. Yellow and green represent the sequences of fecal and BALF samples, respectively. The different colors in the outermost circle represent annotated species. Red triangles represent the sequences present in both the lung and gut samples from the same patient. The phylogenetic analysis was conducted using FastTree 2.1.10 software.

**Figure 6 microorganisms-11-00546-f006:**
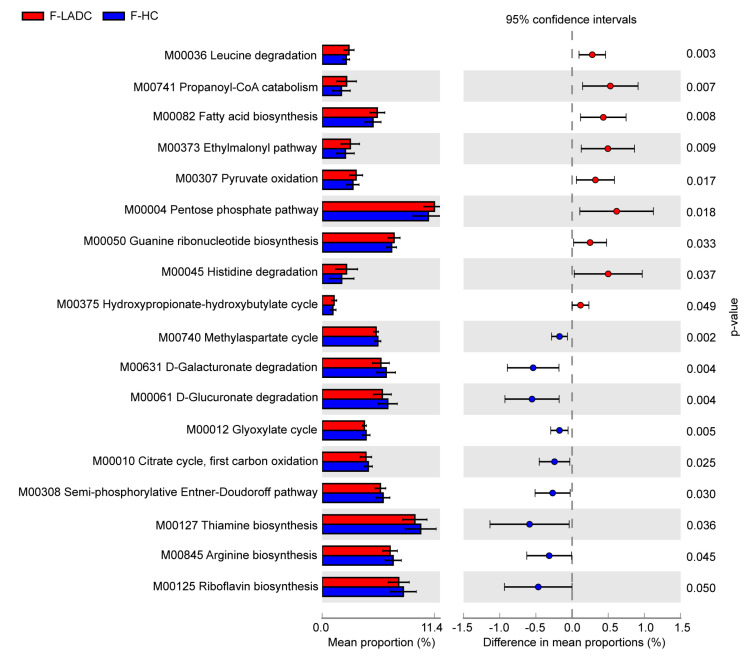
KEGG modules that differ significantly in relative abundance between the gut samples of patients with LADC and healthy controls. The statistical analysis was performed and visualized using the STAMP software. The mean proportion and the difference in the mean proportions for pathways showing a significant difference in abundance are shown. The 95% confidence intervals and statistical significance (*p* < 0.05) are also indicated. Red and blue represent the patients with LADC (n = 43) and healthy controls (n = 64), respectively. A two-sided Welch’s *t* test was used to identify the significantly different metabolic pathways between the two groups, with *p* < 0.05 considered significant.

**Figure 7 microorganisms-11-00546-f007:**
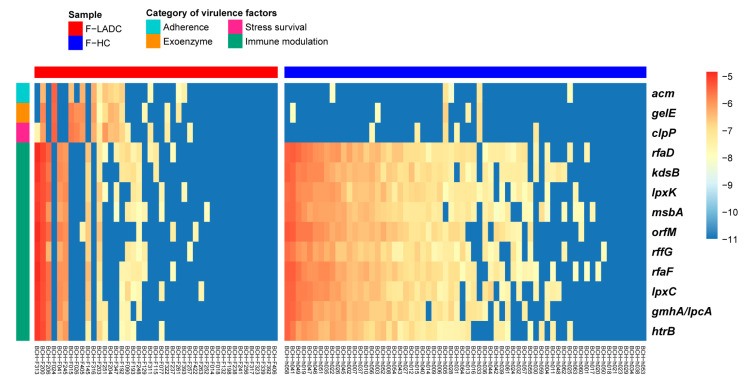
Heatmap of virulence factors with the most differences in abundance from LEfSe analysis. LefSe analysis was performed with *p* < 0.05 and LDA score > 2.5 for the relative abundance of virulence factors. Blue represents lower abundance and red represents higher abundance. The bar at the top indicates sample groups, with red and blue representing patients with LADC (n = 43) and healthy controls (n = 64), respectively. The bar on the left side of the heatmap indicates the category of virulence factors.

**Table 1 microorganisms-11-00546-t001:** Clinical characteristics of patients with LADC and healthy individuals.

Characteristic	Total	LADC	Healthy Controls	*p* Value *
	(n = 107)	(n = 43)	(n = 64)	
**Sample type**				
Feces	107	43	64	
BALF	42	42	NA	
**Sex**				0.5538
Female	50 (46.7)	22 (51.2)	28 (43.8)	
Male	57 (53.3)	21 (48.8)	36 (56.2)	
**Age (years)**				<0.001
Mean (±SD)	50 (±13)	57 (±11)	46 (±12)	
**Smoking status**				0.8345
Current or recent smoker	34 (31.8)	13 (30.2)	21 (32.8)	
Never smoked	73 (68.2)	30 (69.8)	43 (67.2)	
**Disease stage**				
I	20 (46.5)	20 (46.5)	NA	
II	3 (7.0)	3 (7.0)	NA	
III	10 (23.3)	10 (23.3)	NA	
IV	10 (23.3)	10 (23.3)	NA	
**Distant metastasis**				
Absence	33 (76.7)	33 (76.7)	NA	
Presence	10 (23.3)	10 (23.3)	NA	

Data are presented as mean (±SD) for continuous variables and total number (percentage) for categorical variables. * *p* value for categorical variables are based on Fisher’s exsact test, while *p* values for continuous variables are based on Student *t*-test. NA indicates not applicable.

## Data Availability

Raw Sequencing data have been deposited to the NCBI SRA project under the NCBI BioProject ID PRJNA906201.

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
