# Peer review of "Association Studies on Gut and Lung Microbiomes in Patients with Lung Adenocarcinoma"

_microorganisms, 2023, doi:10.3390/microorganisms11030546_

Round 1
Reviewer 2 Report
The study, described by the authors, stimulates interest, however I believe it is appropriate to underline some details.
The authors focused on gut and lung microbiomes in association with Lung adenocarcinoma (LADC).
They used both the fecal and bronchoalveolar lavage fluid (BALF) samples to analyze the bacterial species that were correlated with the development of LADC and also the shared bacterial components of fecal and BALF microbiota, and the metabolic pathways of gut microbiota associated with LADC development.
When abbreviations are used, spell out the full word at first mention in the text followed by the abbreviation in the parentheses. Thereafter, use the abbreviation throughout.
Lines 94-95 and lines 104-107, it would be appropriate to put all the data in a table so as to be more easily consultable
Table 1: please check the legend
